# Species-Specific Discrimination of Insect Meals for Aquafeeds by Direct Comparison of Tandem Mass Spectra

**DOI:** 10.3390/ani9050222

**Published:** 2019-05-07

**Authors:** Ikram Belghit, Erik-Jan Lock, Olivier Fumière, Marie-Caroline Lecrenier, Patricia Renard, Marc Dieu, Marc H. G. Berntssen, Magnus Palmblad, Josef D. Rasinger

**Affiliations:** 1Institute of Marine Research, P.O. Box 1870 Nordnes, 5817 Bergen, Norway; Erik-Jan.Lock@hi.no (E.-J.L.); Marc.Berntssen@hi.no (M.H.G.B.); 2Centre Wallon de Recherches agronomiques (CRA-W), 5030 Gembloux, Belgium; o.fumiere@cra.wallonie.be (O.F.); mc.lecrenier@cra.wallonie.be (M.-C.L.); 3University of Namur, rue de Bruxelles 61, B-5000 Namur, Belgium; patsy.renard@unamur.be (P.R.); marc.dieu@unamur.be (M.D.); 4University of Namur, mass spectrometry facility (MaSUN), rue de Bruxelles 61, B-5000 Namur, Belgium; 5Leids Universitair Medisch Centrum, 2316 Leiden, The Netherlands; N.M.Palmblad@lumc.nl

**Keywords:** insect meal, shotgun proteomics, species differentiation, protein quantification

## Abstract

**Simple Summary:**

Aquaculture is amongst the most efficient ways to produce animal protein for human consumption, and this sector is expected to continue to grow worldwide. Inclusion of novel protein sources, like insect meal, may help to mitigate the expected scarcities of feed resources and reduce environmental pressure. However, considered as processed animal protein (PAP), insect meal must comply with the respective legal constraints associated with PAP legislation to guarantee its safety for use as fish feed ingredients. Therefore, there is a need for the development of methods to identify and quantify the species origin of insect-based ingredients in aquafeed. In this study, we propose high-throughput tandem mass spectrometry for the identification and differentiation of 18 different insect meal samples from the species *Hermetia illucens* (8), *Tenebrio molitor* (5), *Alphitobius diaperinus* (3) and *Acheta domesticus* (2). Using high throughput proteomics tools in combination with direct spectral comparison, we were able to differentiate the insect meal samples according to the taxonomic classification of the insect species. Mass spectrometry-based proteomics is a powerful tool for the species-specific discrimination of insect meals for feed formulations.

**Abstract:**

Insect protein has the potential to become a sustainable feed ingredient for the rapidly growing aquaculture industry. In the European Union, insect derived protein is placed under the same legislation as processed animal proteins (PAP). It is therefore of interest to develop methods for regulatory use, which unambiguously identify the species origin of insect-based ingredients. We performed (i) total protein quantification of insect samples using the traditional nitrogen-to-protein conversion factor of 6.25 and the sum of anhydrous amino acids, (ii) quantitative amino acid profiling and (iii) high-throughput tandem mass spectrometry to describe and differentiate 18 different commercial-grade insect meal samples derived from *Hermetia illucens* (8), *Tenebrio molitor* (5), *Alphitobius diaperinus* (3) and *Acheta domesticus* (2). In addition, we investigated and compared different protein extraction and digestion protocols for proteomic analysis. We found that irrespective of sample preparation, shotgun proteomics in combination with direct spectral comparison were able to differentiate insect meal according to their taxonomic classification. The insect specific spectral libraries created in the present work can in future be used to develop more sensitive targeted methods of insect PAP identification and quantification in commercial feed mixtures.

## 1. Introduction

Aquaculture is amongst the most efficient ways to produce animal protein for human consumption, and this sector is expected to continue to grow in the foreseeable future, putting more pressure on the world’s existing protein sources [1]. Insects, which are part of the natural diet of many fish species, represent a novel sustainable protein resource that can be incorporated into future aquafeed production. The nutritional composition of insects has been widely reviewed with an emphasis on fat, essential minerals and vitamins [2,3]. They are rich in protein (varied depending on the species) and have a well-balanced essential amino acid profile, similar to the amino acids of fish meal (FM); the main conclusions being that many insect species can be optimal feedstuff for animals, including fish [4,5,6,7,8]. The most promising insect species for industrial feed production are black soldier fly (*H. illucens*) larvae, common housefly (*Musca domestica*) larvae, house cricket (*A. domesticus*) and yellow mealworm (*T. molitor*) [9,10]. These species have received increasing attention because they potentially valorize many types of organic side-streams (e.g., from food-producing factories) and produce high-quality protein that can be used in aquafeed [11].

According to a recent European Commission regulation (2017/893-24/05/2017), the use of insect meal (IM) from seven different insect species is allowed to be used in aquafeeds. As a consequence, a tremendous increase in investments in this sector has been observed. However, considered a processed animal protein (PAP) in the European Union (EU), insect feed ingredients must comply with the associated legal to guarantee their safe use in fish feed ingredients. Analytical approaches therefore must be developed which allow for an unambiguous detection and identification of white-listed insect species in insect-protein containing feed ingredients. To guarantee such a safe use of PAP, standard operating procedures (SOP) have been established for the control of aquafeeds by the European Union Reference Laboratory for Animal Protein (EURL-AP). These include the application of light microscopy to detect PAP when feed is not supposed to contain PAP and the use of an EURL-AP validated polymerase chain reaction (PCR) based method, which is used for ruminant DNA-detection when the feed is known to contain PAP or blood products, as indicated from the declaration or the labeling [12]. For insects, to date only a few studies investigated the applicability of qPCR for the detection of specific insects in compound feed. For example, Marien et al. [13], developed a qPCR assay for the detection of *H. illucens;* and Debode et al. [14], for the detection of *T. molitor*. Some investigators also looked at the applicability of light microscopy for different commercially available IMs from *H. illucens*, *Gryllus assimilis*, *A. diaperinus, Bombyx mori* and *T. molitor* [15,16] or Fourier Transform Near Infrared spectroscopy approach (FTNIR) for the detection of *T. molitor* and *A. Domesticus* [17]. However, with the fast-growing number of insect species used for the production of insect meals, the detection, identification and differentiation of insect species by qPCR or microscopy remain a challenge. In addition, the current general paucity of information on insect species in molecular reference databases further hampers any efforts to develop targeted assays. 

Proteomic-based methods using (tandem) mass spectrometry (MS) were, in a recent scientific opinion by the European Food Safety Authority (EFSA), identified as promising tools to complement current standard techniques of feed PAP detection [18]. Different laboratories specialized in feed and food safety have been developing promising MS based tools for the species-specific detection, differentiation and quantification of animal proteins [19,20,21,22]. Most of these methods are targeted approaches which are based on the detection of a known peptide or protein of which the sequence information is available [23,24]. However, similarly to qPCR-based methods, the lack of genetic information of many insect species currently impedes the development of targeted MS approaches for insect PAP. 

Therefore, alternative methods, which can be performed independently of genomic information, are needed to overcome these limitations. Recently, direct spectral library comparisons following an approach first described in Palmblad and Deelder [25] were used by Rasinger et al. [12] to differentiate PAP for aquafeeds according to their taxonomic classification. In the current study, a similar proteomics approach was used to differentiate between 18 commercial-grade insect protein meals derived from *H. illucens* (8), *T. molitor* (5), *A. diaperinus* (3) and *A. domesticus* (2). In addition, the insect meal samples were subjected to quantification of total protein and amino acid profiling. 

## 2. Materials and Methods

### 2.1. Sample Material 

In the present work, eighteen different commercial IM samples were selected based on their availability (IM produced by different companies): 8 samples from species of the Diptera order; black soldier fly larvae (BSF) (*H. illucens*), 8 samples from species of the Coeleoptera order, including the yellow mealworm (YW) (*T. molitor*) and the lesser mealworm (LW) (*A. diaperinus*) and 2 samples from the Orthoptera order; house cricket (HC) (*A. domesticus*) (Table 1). 

### 2.2. Amino Acid Analysis 

Amino acid analyses of IM samples were carried out by ultra-performance liquid chromatography (UPLC, Waters Acquity UPLC system) coupled with a UV detector following an accredited method by the Nordic Committee of Food Analysis (NMKL). The method is described in detail elsewhere [5,26,27]; in short, ground samples equivalent of 30–40 mg of protein were hydrolyzed in 6 N HCl at 110 °C for 22 h. Prior to hydrolysis, 3.125 mM Norvaline (Sigma-Aldrich, St. Louis, MO, USA) was added as internal standard, and 0.1 M Dithiothreitol (DTT, Sigma-Aldrich) was added as an antioxidant agent to protect methionine from degradation during acid hydrolysis. For a further protective aid, a layer of N2 gas was put into the flasks for 30 s, and then the flasks were capped immediately. During acid hydrolysis, tryptophan and cysteine were destroyed. After hydrolysis, the samples were cooled in cold water until room temperature was reached and centrifuged in a vacuum centrifuge to complete dryness. After centrifugation, the residues were diluted in deionized water and filtered through a syringe-driven filter. Prior to the instrumental analysis, a derivatization agent (AccQ.Tag^TM^, Waters, Milford, MA, USA) was added to each sample. Finally, amino acids were separated by UPLC (column: Aquity UPLC BEH C18 1.7 μM (2.1 × 100 mm), Waters, flowrate 0.7 mL min^−1^) and results integrated by Empower 3 (Waters). Amino acids were quantified using standards from Thermo Fisher Scientific (product number; 20088 Rockford, IL 61105, USA). Data was analyzed and visualized using Qlucore Omics Explorer version 3.3 (Qlucore AB, Lund, Sweden).

### 2.3. Total Nitrogen 

Total nitrogen (TN) was analyzed according to the Dumas method [28]. Briefly, wet, ground IM samples using a CHNS elemental analyzer (Vario Macro Cube, Elementar Analysensysteme GmbH, Langenselbold, Germany), using helium as carrier gas. The instrument was calibrated with ethylene diamine tetra acetic acid (EDTA) (Leco Corporation, Saint Joseph, MI, USA). Sulfanilamide (Alfa Aesar GmbH & Co, Karlsruhe, Germany) and a standard meat reference material (SMRD 2000, LGC Standards, Teddington, UK) was used as the control sample.

#### Protein Content 

The protein content was quantified using; (i) the N-to protein factor of 6.25 (crude protein, CP) and (ii) as sum of amino acids residues [29] (true protein, TP). The amino acids residues were calculated as follows: $$E_{i}={\mathrm{AA}}_{i}\times \left(\frac{{\mathrm{AA}}_{i}\;\left(\mathrm{MW}\right)-H_{2}O\left(\mathrm{MW}\right)}{{\mathrm{AA}}_{i}\;\left(\mathrm{MW}\right)}\right)$$
where = the proportion of the single amino acid (g amino acid per 100 g of dry weight); MW= molecular weight of a single amino acids. 

Calculation of N-Protein conversion factors (*kP*) was determined for each IM sample as follows:$$\sum \frac{E_{i}}{TN}$$
where $E_{i}$ represents the gram of the single amino acid residue per 100 g of dry weight and TN represents the gram of N per 100 g of dry weight.

### 2.4. Proteomics Analysis 

#### 2.4.1. Extraction, Solubilization and Quantification of Proteins

The proteins were extracted from the IM samples by using two different methods (protein extractions 1 and 2) in parallel in two different laboratories (laboratories A and B), respectively;

##### Protein Extraction 1: 

Fifty (50) mg of IM samples was solubilized with 0.5 or 1 mL of lysis buffer (4% SDS, 0.1 M Tris-HCl, pH 7.6). The samples were ground in tubes containing resin on ice using pestle (Sample Grinding Kit, GE Healthcare Life Science, 80648337, Piscataway, NJ, USA). Then, 1 M Dithiothreitol was added to the extraction solution, to obtain a final concentration of 0.1 M. Tubes were centrifuged for 10 min at maximum speed to remove resin and cellular debris. The supernatants were collected and heated at 95 °C for 5 min and then stored at −20 °C until further use. 

##### Protein Extraction 2: 

Two (2) mL of extraction buffer (200 mM Tris-HCl, pH 9.2, 2M urea) was added to 200 mg of IM samples. Samples were shaken for 30 min at room temperature in a Grant-bio rotating-shaker (Grant instruments Ltd, Camb, England) followed by sonication for 15 min. Tubes were then centrifuged at 14,000 rpm for 10 min at 4 °C and supernatants were collected. The obtained protein extracts samples were stored at −20°C until further use.

At both laboratories, protein concentrations were determined by the Pierce™ 660 nm Protein Assay Reagent (ThermoFisher Scientific, Waltham, MA, USA) using BSA as protein standard (ThermoFisher Scientific).

#### 2.4.2. In-Solution Digestion of Proteins

Insect protein extracts were digested using modified filter-aided sample preparation (FASP) method as described by Wiśniewski et al. [30] and Distler et al. [31]. At laboratory A, 50 µg of the protein extracts were diluted with 200 µL of 8 M urea solution (100 mM Tris-HCl, pH 8.5) and loaded into ultrafiltration spin column (Microcon 30, Millipore, Burlington, MA, USA). Proteins were then alkylated in 50 mM of iodoacetamide (C_2_H_4_INO) for 20 min in darkness at room temperature. After that, the protein mixtures were washed with 200 µL of 8 M urea solution and 100 µL of 50 mM ammonium bicarbonate (NH_4_HCO_3_) solution. Trypsin (1:50, enzyme to protein) was added and incubated with the protein mixture at 37 °C for 16 h. The filter unit was centrifuged and washed with 40 µL of 50 mM ammonium bicarbonate solution and 50 µL of 0.5 M NaCl. The eluents containing tryptic peptides were vacuum dried. At laboratory B, ultrafiltration spin columns (Microcon 30, Millipore) were washed with a 1% formic acid solution and 40 µg of protein extracts were added to the membrane. After washing the filter with 8 M urea in 100 mM Tris-HCl (pH 8.5), reduction and alkylation were conducted on the filter devices at 56 °C and room temperature, respectively. Excess of iodoacetamide was removed by an additional DTT incubation followed by a washing step with 50 mM ammonium bicarbonate buffer. Then, 800 ng of trypsin was added to the membrane and incubated overnight at 37 °C for 16 h. The next day, peptides were eluted by centrifugation at 11,000 g and a second elution was done with 50 mM ammonium bicarbonate buffer. Samples were concentrated using a vacuum centrifuge.

Following trypsin digestions, at both laboratories, peptide concentrations were determined by adsorption at 280 nm wavelength using a Nanodrop 2000 (Thermo Scientific). Peptide samples (laboratories A and B) were stored at −20 °C prior to mass spectrometry analysis.

#### 2.4.3. Mass Spectrometry 

All the proteins digest (extractions 1 and 2) were analyzed using an ESI-MS/MS maXis Impact UHR-TOF (Bruker, Billerica, MA, USA) coupled with an UltiMate 3000 HPLC system (Thermo Scientific) (chromatogram is presented in Appendix A). The digests were separated by reverse-phase liquid chromatography using a 1 mm I.D. × 150 mm reverse phase column (Acclaim PepMap 100 C18, Thermo Scientific). The flow rate was 40 µL/min. Mobile phase A was 95% water, 5% acetonitrile, 0.1% formic acid. Mobile phase B was 20% water, 80% acetonitrile, 0.1% formic acid. The digest was injected and the organic content of the mobile phase was increased linearly from 4% B to 40% B in 60 min and from 40% B to 90% B in 10 min, and then washed with 90% B for 10 min and with 4% B for 10 min, for a total of 90 min. The column effluent was directly connected to the MS. In survey scan, MS spectra were acquired for 0.5 sec in the mass to charge (m/z) range between 50–2200. The most intense peptides ions 2+ to 4+ were sequenced during a cycle time of 3 sec. The collision-induced dissociation (CID) energy was automatically set according to m/z ratio and charge state of the precursor ion. The mass spectrometer and HPLC systems are controlled by Compass Hystar 3.2 (Bruker).

### 2.5. Proteomics Bioinformatics and Data Mining

Mass spectrometry data generated using the UHR-TOF (.baf files) were converted using compassXport (Bruker DataAnalysis 4.2) and saved as mgf and mzXML files. Standard bottom-up proteomics bioinformatics analysis was conducted using the proteoQC package (version 1.18.1) [32] in R (version 3.4.4) [33] running in RStudio (version 1.0.143) [34]. In short, mgf peak lists obtained from MS/MS spectra were subjected to the X!Tandem search engine [35] with the following parameter settings: (i) Trypsin was set as digestive enzyme and a maximum of two missed cleavages were allowed, (ii) MS1 and MS2 tolerances were set to 10.0 ppm and 0.05 Da respectively, and (iii) carbamidomethylation of carbon and oxidation of methionine were set as fixed and variable modifications, respectively. The peak lists of BSF, YW, LW and HC were searched individually against their respective UniprotKB/Swiss-Prot (release 2019_01) reference proteomes (Table 2). Protein identifications were inferred from peptide identifications; each identified protein had at least one associated unique peptide sequence identified at q-value equal or less than 0.01 (equivalent to a 1% FDR). The Occam’s razor approach [36] was applied to deal with degenerate peptides by finding a minimum subset of proteins that covered all of the identified peptides. Insect specific proteins and peptides detected across both laboratories and analyses are reported (Online resource). To compare LC-MS/MS datasets directly, independently of sequence databases, DISMS2 was used, a spectral library comparison pipeline frequently used to calculate proteome-wide distances between samples without suitable peptide/protein reference databases [37]. Venn diagrams comparing the numbers of peptides and proteins are presented with an interactive tool for comparing lists with Venn’s diagrams [38].

## 3. Results and Discussion

### 3.1. Quantification of Total Protein and Amino Acid Profiling

Insect meal has been highlighted as a suitable replacement of FM and plant-based diets, such as soy protein, in fish diets [2,3,5,8]. However, when novel proteic feed components are introduced, the amino acid (AA) composition should be well balanced in order to meet the requirements of fish species. The AA profile of IM samples studied in the current study were comparable to those given in previous reports for BSF, YW, LW and HC [39,40] (Table 3). Moreover, the IM investigated contained all the essential AA and in quantities that are similar or even higher than FM and soy protein, except for lysine and methionine (Table 3). In aquafeed, these two AAs are the main limiting essential AAs and are generally supplemented in the plant-based diets in order to fulfil the requirement of carnivorous fish species [41,42]. Among the four insect species analyzed in the current study, only HC and LW meals were found to contain taurine; in BSF and YW meals taurine levels were not detected (Table 3). It was also found that the amount of taurine was much higher in IM samples derived from HC than LW species (0.8–1% and 0.2–0.3% of crude protein, respectively) (Table 3). Similar results have been reported by Finke [43], where HC contained substantial level of taurine when compared to other insect species (*zophobas morio*, *T. molitor*, *galleria mellonella* and *lumbricus terresstris*). In the present study, the amount of taurine detected in HC meals (~5 g/kg wet basis) was higher even than the levels reported by Finke (0.18 mg/g wet basis) (Appendix A) [43,44]. Similarly to a typical herring FM, which has taurine level between 0.5–2% of crude protein, the HC meal might therefore be a suitable source of taurine in future fish feed formulations [45]. 

The AA composition of the eighteen IM samples studied varied between the species (Table 3). A principal component analysis (PCA) was applied to the AAs data and resulted in a clear distinction between the amino acids profile of HC, BSF, LW and YW meals, respectively (Figure 1a). Euclidean distance clustering confirmed these patterns, as demonstrated by the heat maps in Figure 1b. In addition, a clustering of amino acid profiles was observed which reflected the taxonomic classification of the insect species; Diptera, Coleoptera and Orthoptera (Figure 1b). IM derived from BSF species, however, was separated into two different groups based on the AA profile (Figure 1b). This observation might be related to the different processing steps the IM was subjected to, which differed between the commercial companies supplying the samples. For example, some samples were labeled as defatted; a process which increases the crude protein content and consequently the AAs composition of IM from different insect species [40,48]. However, since no further information was provided about the processing steps performed by the different IM producers, its effects on AA composition remains to be investigated. 

The protein content of the eighteen IM samples is shown as both crude protein (based on total nitrogen, calculated by using nitrogen-to-protein conversion factor of 6.25) and true protein (sum of anhydrous AAs) in Table 4. Crude protein values of the IM samples were found to be much higher than the true protein amounts with an overestimation of the protein content ranging between 20–28% (Table 4). This overestimation by using the standard 6.25 N-to-protein factor is due to the presence of the non-protein nitrogen in insect species. Janssen et al. [39], calculated the specific N-protein conversion factor for BSF, YW and LW species and found a *kP* value of 4.76. Similarly, in the current study, the *kP* values calculated for BSF, YW, LW and HC meals varied between 4.21–5.01, 4.64–4.86, 4.83–5.0 and 4.53–4.80, respectively (Table 4). In other words, the nitrogen-to-protein conversion factors calculated in the present study are in line with data reported in the literature [27,39] and provide for a more correct estimate of the IM protein content which is a vital determinant when insects are to replace current protein sources in aquafeed. 

### 3.2. Proteomics Analysis of Insect Meal Samples

Protein extraction and solubilization are key steps for protein identification in bottom-up proteomics [49]. The protein concentration and yield obtained from the two different methods used to extract protein from IM samples are presented in Appendix A. The protein extraction efficiencies varied largely between insect samples and also between the two extraction methods. In both methods tested, the protein recoveries were higher in the samples extracted from LW (20–58% and 5–13%, extraction 1 and 2, respectively) than that from BSF, YW and HC species (Appendix A). The relative protein yield obtained by using a 4% SDS and 0.1 M Tris-HCl, pH 7.6 based buffer (extraction 1) in general was higher for all investigated IMs samples than the one obtained using a 200 mM Tris-HCl, pH 9.2, 2 M urea-based buffer (extraction 2; Appendix A). These differences in protein concentration between the two extraction methods might indicate a possible effect of the different steps and reagents used for protein extraction. In extraction 1, samples were ground to fine powder. This increases the surface to volume ratio and facilitates the permeation of the sample with homogenization buffer achieving a higher extraction yield [49]. In addition, in extraction 1, SDS was used, which is known as an anionic detergent with superior solubilization power. These factors might in part explain the higher yield obtained for extraction 1. However, the large degree of variability between both samples and methods used calls for additional experiments to optimize and standardize extractions procedure since protein isolation is the first critical step for proteomic studies [50].

In earlier work on PAP we demonstrated that data mining techniques can successfully be applied to bottom-up shotgun mass spectrometry data to obtain robust and reliable species-specific peptide markers [12]. However, this approach requires common proteins to be identified across all species of interest [23]. In the present work, the mass spectra obtained were analyzed using proteoQC [51] (mass spectra of all samples are available online), a total of 1360 peptides and 173 insect specific proteins were identified (see Appendix A and MassIVE dataset [52]). Only 269 peptides and 86 proteins were detected consistently across the two laboratories (see Figure 2, Appendix A and MassIVE dataset [52]). As can be seen in Table 5, the bulk of these proteins were identified in samples derived from YW; only a handful of proteins were identified in samples derived from BSF and HC and no proteins were identified in samples derived from LW (Table 5). As was observed previously, the rate of PAP protein identification is mainly related to the size of the UniprotKB/Swiss-Prot databases [51].

Despite the importance of insects for agricultural ecosystems, many insect species lack up-to-date genomic data [53], which is a challenge for the development and application of insect specific molecular tools. When there is a paucity of genomics information, direct comparisons of mass spectra can be performed [37]. We recently demonstrated that using compareMS2, an approach for the direct pairwise comparison of tandem mass spectra [25], allowed for the species and tissue specific differentiation of PAP of bovine, ovine, porcine and avian origin [12]. Contrary to the identification of peptides and proteins using database dependent search algorithms, tools like compareMS2 and the recently published DISMS2 [37] make use of the bulk of high quality tandem mass spectra rather than relying on selected few. As can be seen in Table 5, when applying a peptide mass fingerprinting approach, only a fraction of the detected spectra (tSpectra) was used in the identification and differentiation of IM peptides (iSpectra). In the current study, we present two representative dendrograms of DISMS2, a pipeline used to calculate distance matrices created from tandem mass spectra (Figure 3). All mass spectra of samples from extraction 1 were used and successfully arranged the 18 samples according to the insect species they originated from (Figure 3). In addition, in accordance with to the amino acid profile data (Figure 1b), also the spectral library clustering reflected the taxonomic classification of the insect order (Diptera, Coleoptera and Orthoptera). Phylogenetic analysis of data obtained from extraction 2, also correctly arranged the 18 samples according to the taxonomic classification; however, with two exceptions (BSF6 and YW13). This difference in clustering observed could be due to the different extraction methods used and highlights once again that an optimization and standardization of the protein extraction procedures is critical for proteomics analyses [50]. In addition to sample preparation the application of filter settings of the DISMS2 algorithms can be optimized to yield best results in specific situations [37]. Therefore, optimization of both sample preparation and algorithm settings should be investigated further to allow for the creation of high-quality PAP spectral library reference collections suitable for differentiation of insect PAP in feed mixes. Possibly, such library collections could in the future also be used for quantification of insect proteins in feed mixes as recently demonstrated for raw and processed fish and meat samples [20,54,55]. Additionally, spectral libraries can be mined for suitable signature spectra, which can be used for the generation of specific targeted mass spectrometry assays for absolute quantification of PAP in feed mixtures at legal limits of below 0.1% (w/w).

## 4. Conclusions

In conclusion, our results confirmed the suitability of insect meals as a source of proteins for future fish feed formulations. A nitrogen-to-protein conversion factor of ~4.21–5.0 may be more suitable than the usual factor of 6.25, when quantifying total protein content of insect meals derived from BSF, YM, LW and HC. The different sample preparation approaches employed in the present work strongly affected total protein yield and identification rates in bottom-up proteomics. To a lesser extent also the direct spectral comparisons were affected by the different protein extraction and digestion procedures employed; yet the DISMS2 algorithm was able to successfully differentiate insect meals according to their taxonomic groups. It is therefore of interest to further explore the use of database agnostic approaches for regulatory uses which aim for the identification and differentiation of the species origin of insect-based feed ingredients. Future work will focus on the standardization of sample preparation procedures and the creation of an extensive high quality and freely available insect PAP spectral library reference collection, which can be used for species identification by spectral library matching, as exemplified in [56] and for the development of highly sensitive targeted mass spectrometry assays, respectively.

## Figures and Tables

**Figure 1 animals-09-00222-f001:**
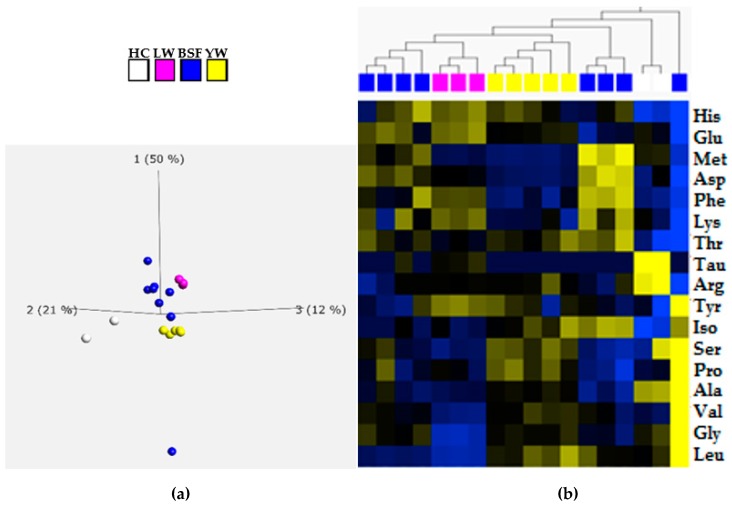
Principal analysis (PCA) and hierarchical clustering analysis of the amino acids content from 18 IM samples: (**a**) The plot shows the three components, PC1 (50%), PC2 (21%) and PC3 (12%). The white, pink, blue and yellow dots represent house cricket (HC), lesser mealworm (LW), black solider fly (BSF) and yellow mealworm (YW), respectively; (**b**) Semi-quantitative visual heat map representation of amino acids content in the 18 IM samples. The white, pink, blue and yellow square represent house cricket (HC), lesser mealworm (LW), black solider fly (BSF) and yellow mealworm (YW), respectively. Each line in the heat map represents the content of amino acids. The deeper the yellow color, the higher is the amino acid content in the respective sample; similarly, the deeper the blue color, the lower is the amino acids content in the respective sample.

**Figure 2 animals-09-00222-f002:**
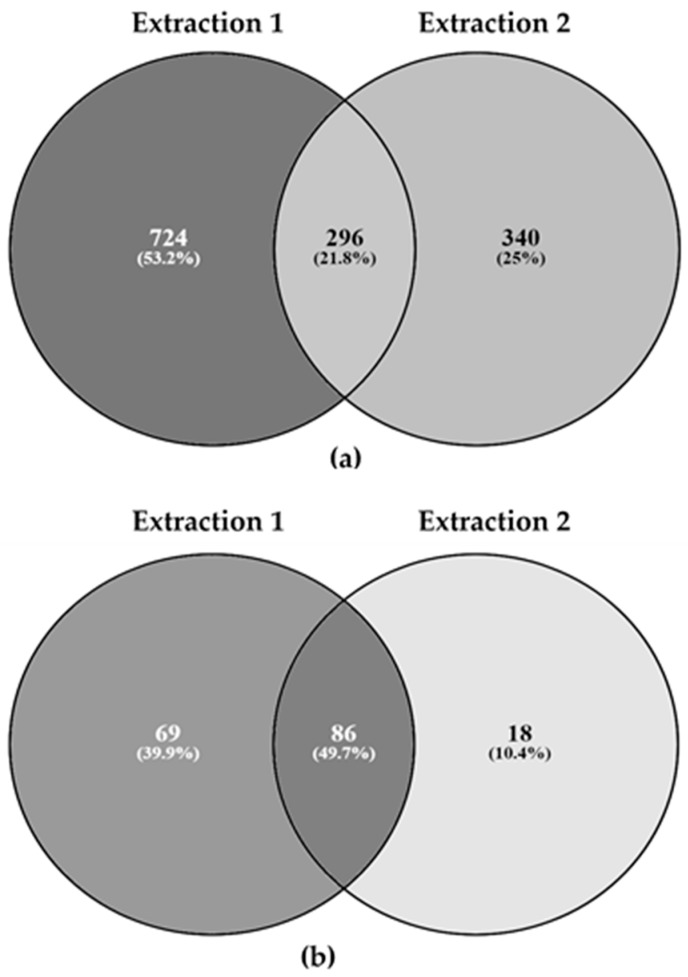
Venn diagrams comparing the numbers of peptides (**a**) and proteins (**b**) detected in 18 IM samples using a bottom-up proteomics. Extraction 1 and 2 refer to the two different extraction methods applied in Laboratory A and B. The intersection of the two circles shows the number of commonly detected peptides (**a**) and proteins (**b**).

**Figure 3 animals-09-00222-f003:**
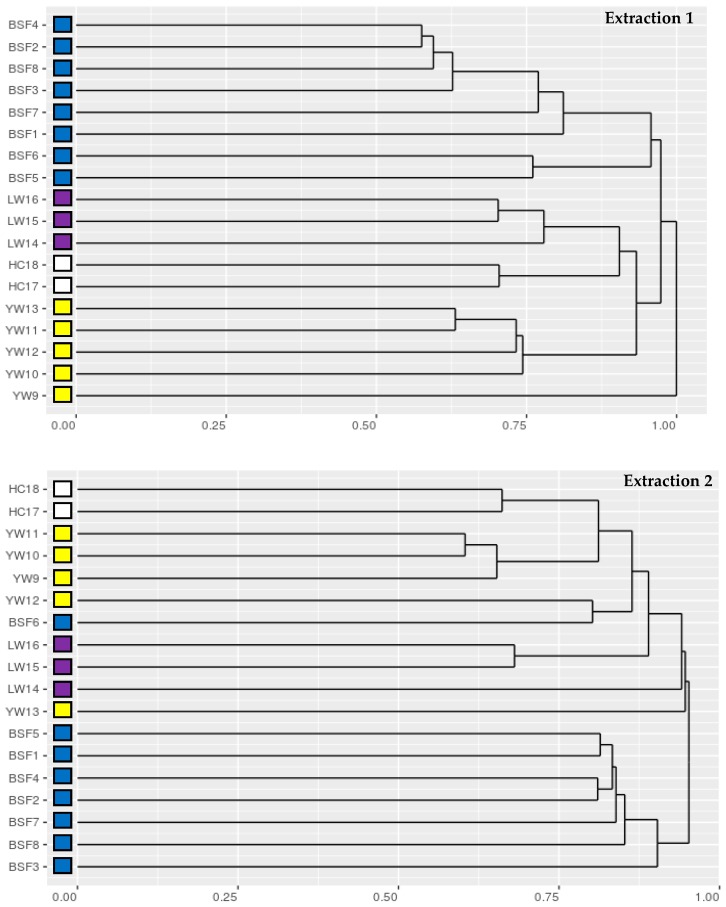
Species specific insect meal samples differentiation. Direct comparison of spectra obtained by tandem mass spectrometry usingDISMS2. With two exceptions (BSF6 and YW13 in Laboratory B) insect meal samples cluster according to the taxonomic classification of the insect species.

**Table 1 animals-09-00222-t001:** Insect protein samples included in the study, with the *Latin name*, order and the family belongings.

Samples	Species	Latin Name	Order-Family
**BSF1-BSF8**	Black soldier fly	*Hermetia illucens*	Diptera- Stratiomyidae
**YW9-YW13**	Yellow mealworm	*Tenebrio molitor*	Coleoptera-Tenebrionidae
**LW14-LW16**	Lesser mealworm	*Alphitobius diapernius*	Coleoptera-Tenebrionidae
**HC17-HC18**	House cricket	*Acheta domesticus*	Orthoptera- Gryllidae

BSF = black soldier fly; YW = yellow mealworm; LW = lesser mealworm; HC = house cricket.

**Table 2 animals-09-00222-t002:** Taxon identifier and number of proteins of *H. illucens, T. molitor*, *A. diaperinus* and *A. domesticus*.

Species	Taxon Identifier	Number of Proteins (UNIPROT)
*H. illucens*	343,691	71 (1)
*T. molitor*	7067	532 (26)
*A. diaperinus*	27,448	28 (0)
*A. domesticus*	6997	131 (4)

Number of predicted and (reviewed) proteins of *H. illucens, T. molitor*, *A. diaperinus* and *A. domesticus* listed in the UniprotKB/Swiss-Prot reference proteome database.

**Table 3 animals-09-00222-t003:** Total amino acid composition (% of crude protein) of the 18 insect meal samples.

Species	Ala	Arg	Asp	Glu	Gly	His	Ile	Leu	Lys	Met	Phe	Pro	Ser	Tau	Thr	Tyr	Val
**BSF 1**	9.5	6.2	12.4	16.0	6.7	3.8	5.2	8.8	6.9	2.0	5.2	8.8	6.0	<LoD	5.2	8.1	7.4
**BSF 2**	7.7	6.8	12.3	14.2	7.1	4.2	5.2	8.7	7.7	2.4	6.5	7.7	5.5	<LoD	5.2	9.4	7.1
**BSF 3**	8.1	5.6	12.6	15.2	6.7	3.3	5.2	8.5	8.1	2.1	5.6	7.4	5.6	<LoD	5.2	8.1	7.4
**BSF 4**	8.2	6.7	12.4	15.5	6.7	3.9	5.5	8.8	8.8	2.1	5.2	7.0	5.5	<LoD	5.2	7.9	7.0
**BSF 5**	8.1	6.1	13.9	13.9	6.4	3.6	5.6	9.4	8.1	2.6	6.4	6.9	5.3	<LoD	5.3	8.6	7.2
**BSF 6**	13.1	4.5	8.6	10.0	8.6	2.8	5.5	10.3	5.2	1.4	4.1	10.7	6.6	<LoD	4.5	10.7	9.3
**BSF 7**	7.1	6.9	13.6	13.8	6.0	3.8	5.5	9.3	9.0	2.7	6.4	6.2	5.2	<LoD	5.5	7.9	6.7
**BSF 8**	7.3	6.4	13.2	12.7	6.4	3.4	5.5	9.5	9.1	2.7	6.4	7.0	5.5	<LoD	5.2	8.9	7.3
**YW 9**	9.0	6.9	10.8	14.7	6.7	3.7	5.5	9.6	7.5	1.7	5.1	8.2	5.9	<LoD	5.1	9.0	7.6
**YW 10**	9.2	6.7	10.8	14.6	6.7	3.8	5.4	9.2	7.4	1.7	4.6	8.7	5.9	<LoD	5.1	9.0	7.2
**YW 11**	9.2	6.8	10.8	14.5	6.6	3.7	5.3	9.2	7.4	1.7	4.7	8.7	6.1	<LoD	5.3	9.5	7.4
**YW 12**	9.0	6.7	11.0	15.0	6.9	3.3	5.6	10.0	6.7	1.8	5.2	8.1	6.0	<LoD	5.4	8.8	7.5
**YW 13**	9.5	7.6	10.8	15.1	6.8	3.8	5.4	9.7	8.1	1.7	4.6	8.6	6.2	<LoD	5.4	7.6	7.6
**LW 14**	8.1	6.7	11.4	16.0	5.6	4.0	5.1	8.4	8.6	1.8	5.6	7.2	5.3	0.3	5.1	9.3	6.5
**LW 15**	8.1	6.7	11.6	15.8	5.6	4.0	5.1	8.4	8.4	1.8	5.8	7.7	5.3	0.2	5.1	9.8	6.7
**LW 16**	8.1	6.7	11.9	15.8	5.6	4.0	5.3	8.6	8.8	1.8	5.8	7.9	5.3	0.2	5.1	9.8	6.7
**HC 17**	11.1	8.7	11.7	14.3	7.0	3.0	5.0	9.3	7.2	2.1	4.6	7.6	6.5	1.0	4.8	6.5	6.7
**HC 18**	10.7	8.3	10.5	14.9	6.8	2.9	4.9	9.3	7.8	2.1	4.6	7.3	5.4	0.8	5.1	8.5	7.1
**FM ***	6.1	5.2	9.3	13.1	6.6	2.1	5.1	6.5	10.1	3.1	3.7	4.2	4.4	1.0	3.5	2.9	4.5
**SP ***	3.7	5.1	11.5	20.7	3.7	1.3	4.0	6.1	3.7	1.4	3.4	5.1	5.3	-	3.7	3.1	4.5

BSF = black soldier fly; YW = yellow mealworm; LW = lesser mealworm; HC = house cricket; FM = fish meal; SP = soy protein; <LoD = < minor than Limit of Detection;* [46,47].

**Table 4 animals-09-00222-t004:** Crude protein (CP, expressed as percentage of wet weight) and true protein (TP) content of 18 insect meal samples, and difference between CP and TP (∆ %) and N-Prot conversion factors.

Abbr	CP	TP	∆%	N-Prot Factor
**BSF1**	72	42	28	4.51
**BSF2**	76	31	24	4.74
**BSF3**	73	27	27	4.59
**BSF4**	73	33	27	4.55
**BSF5**	78	36	22	4.85
**BSF6**	67	29	33	4.21
**BSF7**	78	42	21	4.91
**BSF8**	80	44	20	5.01
**YW9**	76	51	24	4.73
**YW10**	75	39	25	4.67
**YW11**	77	38	23	4.81
**YW12**	78	52	22	4.86
**YW13**	74	37	26	4.64
**LW14**	78	43	22	4.89
**LW15**	81	43	19	5.05
**LW16**	80	43	20	4.98
**HC 17**	73	46	27	4.53
**HC18**	77	41	23	4.80

Protein content of the insect meal samples is presented both as crude protein (calculated as the total nitrogen, using the nitrogen-to-protein factor of 6.25 and true protein (calculated as sum of amino acids residues) (for more details, see material and methods section); TP = true protein; CP = crude protein; ∆% = difference between CP and TP in %; BSF = black soldier fly; YW = yellow mealworm; LW = lesser mealworm; HC = house cricket.

**Table 5 animals-09-00222-t005:** Total numbers of spectra and successfully identified insect specific proteins and peptides in 18 insect meal samples, performed independently at two different laboratories (A and B).

Species	Extraction 1				Extraction 2			
	tSpectra	iSpectra	Peptides	Proteins	tSpectra	iSpectra	Peptides	Proteins
**BSF1**	14,656	4	4	4	16,235	4	4	1
**BSF2**	14,620	35	23	10	15,712	13	8	2
**BSF3**	14,182	6	5	4	15,724	12	11	1
**BSF4**	14,514	21	15	9	16,616	5	4	1
**BSF5**	11,671	25	14	10	15,992	5	4	2
**BSF6**	11,856	73	23	7	11,909	4	1	1
**BSF7**	13,527	19	9	7	14,608	7	5	4
**BSF8**	14,499	18	14	8	16,458	7	7	3
**YW9**	17,416	1	1	1	15,469	1483	365	54
**YW10**	13,671	1422	406	84	14,551	1256	304	62
**YW11**	14,343	1987	548	103	11,778	4	2	2
**YW12**	13,920	2138	431	69	12,986	758	140	33
**YW13**	15,042	1627	524	92	16,837	958	414	72
**LW14**	13,955	0	0	0	15,074	0	0	0
**LW15**	12,654	0	0	0	14,267	0	0	0
**LW16**	13,341	0	0	0	14,392	0	0	0
**HC17**	13,732	39	16	5	15,327	132	22	3
**HC18**	13,131	33	11	6	14,520	83	14	2

tSpectra = total spectra determined; iSpectra = spectra identified; BSF = black soldier fly; YW = yellow mealworm; LW = lesser mealworm; HC = house cricket.

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
