# Peer review of "Species-Specific Discrimination of Insect Meals for Aquafeeds by Direct Comparison of Tandem Mass Spectra"

_animals, 2019, doi:10.3390/ani9050222_

Round 1

Reviewer 1 Report

The authors are here presenting a MS/MS based approach for the discrimination for insect meals for aquafeeds. The topic is interesting because it deals with themes related to sustainable aquaculture and the work brings new knowledge to the subject. 

My general consideration is that the manuscript is well written, correctly structured and the type of research proposed is innovative in the context of the analytical control of animal feed. 

Neverthless, some parts regarding the analytical procedures adopted must be enhanced/clarified. More specifically, some aspects of the determination of amino acids are not clear. 

Are the authors using a method provided by literature? If this is the case, please provide a reference

Is the method instead developed by the authors? If this is the case, please provide a chromatogram and also address all the aspect regarding the method performances (Linearity, precision etc.)

The hydrolysis is performed at 110 °C. At this temperature, also Maillard reaction (which involves  sugars and amino acids) takes place. Can the authors address this aspect?

How was the quantification performed? External calibration, internal standard method....Please specify. If the content of the amino acids is expressed only with % area, this should not be called "quantification". 

Table 3 should also report standard deviations. 

In the TAU column, the authors report "LOQ" for many of the samples. Does it mean that values are lower than limit of quantification? How was this calculated?

Author Response

Comments and Suggestions for Authors

The authors are here presenting a MS/MS based approach for the discrimination for insect meals for aquafeeds. The topic is interesting because it deals with themes related to sustainable aquaculture and the work brings new knowledge to the subject. 

My general consideration is that the manuscript is well written, correctly structured and the type of research proposed is innovative in the context of the analytical control of animal feed. 

Neverthless, some parts regarding the analytical procedures adopted must be enhanced/clarified. More specifically, some aspects of the determination of amino acids are not clear. 

Are the authors using a method provided by literature? If this is the case, please provide a reference

Yes, in this experiment we used an accredited method by our institute (Institute of Marine research, Bergen, Norway) and by the Nordic Committee of Food Analysis (NMKL). We clarified and add description in the material and method section and added references:

Please see below the method references;

S.A Cohen and D.P michaud, Anal.Biochem(1993), 211, 279-287 15.2 15.3 15.4 15.5

Hong-Ji Liu, J.Chromatography (1994), 610, 59-66

S.A Cohen and K.M De Antonis, J.Chromatography(1994), 661, 25-34

Waters, AccQ-TagTM Method. 715001320, REV D.

Is the method instead developed by the authors? If this is the case, please provide a chromatogram and also address all the aspect regarding the method performances (Linearity, precision etc.)

Please see below the method performances:

VALIDATION PARAMETERS

Specificity: There is no peaks within Reagent Blank has no peaks which interferes with the amino acid peaks. The method has good specifity.

Linear range: Hydrolyzed Amino Acids provides good linearity between 25pmol/µl - 1000pmol/µl.

LOD and LOQ:

Trueness: Trueness is determined by analyzing reference material SRM 1849 and CRM 2387. Trueness for both matrixes is between 80 – 120 % for all the amino acids analyzed.

Trueness by SLP: We have analyzed the following matrixes: protein powder, chicken feed, baby food, food supplement and wheat. The result of those analyzes gives a trueness of 80 – 120 % except Tyrosine which is 70 % as the lowest.

Precision: Analysis of granulated Cod shows satisfactory results for all amino acids due to reproducibility. See table.

Internal reproducibility: Internal reproducibility is determined in fish feed, feces, protein powder, chicken feed, peanut butter, granulated Cod, fish balls, pig meat, whole wheat meal and soy.

See example for protein powder:

We provided information about the chromatogram of the standard and the sample, please see below:

The hydrolysis is performed at 110 °C. At this temperature, also Maillard reaction (which involves sugars and amino acids) takes place. Can the authors address this aspect?

We agree with this comment, that high temperatures might promote Maillard reaction between amino acid and sugars, however this reaction happen when the hydrolysis occurs on dry heat process, nevertheless in our method the hydrolysis was performed on wet heat process, which will not trigger a Maillard reaction.

We didn’t have any problem with Maillard reaction in our previous samples and we used this accredited method in our lab and it has been published in several papers:

Espe, M.; Andersen, SM.; Holen, E.; Rønnestad, V.; Veiseth-Kent, E.; Zerrahn, JE.; Aksnes, A. Methionine deficiency does not increase polyamine turnover through depletion of hepatic S-denosylmethionine. Br.J. Nutr. 2014, 112, 1274-1285.

Liland, NS.; Biancarosa, I.; Araujo, P.; Biemans, D.; Bruckner, CG.; Waagbø, R.; Torstensen, BE.; Lock, E-J. Modulation of nutrient composition of black soldier fly (Hermetia illucens) larvae by feeding seaweed-enriched media. PLOS ONE. 2017, 12, 0183188.

Biancarosa, I.; Espe, M.; Bruckner, CG.; Heesch, S.; Liland, N.; Waagbø, R.; Torstensen, B.; Lock, EJ. Amino acid composition, protein content, and nitrogen-to-protein conversion factors of 21 seaweed species from Norwegian waters. J. Appl. Phycol. 2016, 1-9.

Belghit, I.; Liland, NS.; Waagbø, R.; Biancarosa, I.; Pelusio, N.; Li, Y.; Krogdahl, Å.; Lock E-J. Potential of insect-based diets for Atlantic salmon (Salmo salar). Aquacul. 2018, 491, 72-81.

Furthermore, the method has been approved by the Nordic Committee of Food Analysis (NMKL), which is accreditation authority for the Nordic countries:

https://www.nmkl.org/index.php/en/about-nmkl

NMKL-PROSEDYRE NR 3 (1996).   Kontrollkort og kontrollprøve.

NMKL-PROSEDYRE NR 4 (1996 og 2009).   Validering av kjemiske analysemetoder.

NMKL-PROSEDYRE NR 5 (1997).   Måleusikkerhet.

NMKL-PROSEDYRE NR 9 (2007).   Referansematerialer.

How was the quantification performed? External calibration, internal standard method....Please specify. If the content of the amino acids is expressed only with % area, this should not be called "quantification". 

The quantification was performed by internal standards and external standard regression;

Use the internal standard method. The result is calculated by Empower as follows:

The area of the peaks is measured both for standard and sample. The response is corrected due to the response the internal standard (is).

R= Area (aa)* C(is)/ Area (is)

Empower calculates a standard curve for each Amino Acid using linear regression.

y =bx+a

a= 0, the curve is forced through the origo.

y= R

x= C2

b= response factor for each component.

C1=(C2*V)/w

R= Corrected response for AA, mg/ml

Area (AA) = Response for AA (area in AU)

Area (is) = Response for internal standard, (area in AU)

C(is)= Concentration of internal standard, mg/ml

C2= Concentration of the injected sample, mg/ml

V= Dilution, ml

w= Weight of the sample, g

The following parameters must be put into Empower:

- Concentration of all component in standard, mg/ml - Concentration of internal standard in all samples, mg/ml

C (is) = (3,0313 µmol/ml x 117,15µg/µmol x 0,5 ml)/ (5ml x 1000ml/l)  = 0,0355 mg/ml

- w, g sample

- V, Dilution, ml

Table 3 should also report standard deviations. 

The objective of this study was to develop a method for regulatory use, able to differentiate insect meal originated from different insect species. The samples were randomly provided by different companies based on their availability in the market, thus we could have only 1 sample from each company and in this study was not possible to have duplicate samples.    

In addition, in this method we have a higher precision and reproducibility. We analyzed the reproducibility of 9-10 samples for different samples and the standard deviation varied between

 0.1-1.5.

Please see below some examples for the analyzed samples with SD and % RSD:

In the TAU column, the authors report "LOQ" for many of the samples. Does it mean that values are lower than limit of quantification? How was this calculated?’

Sorry for this mistake, in those samples, Tau concentration was not detected, thus the reported LOQ was replaced with not detected (N.D).

Please see the table for the range of quantification for the different AA.

In addition, we presented the AA concentration both as % of crude protein (Table 3) and in mg/g of wet weight (supplementary Table 1).

Reviewer 2 Report

 Authors need to make the following corrections:

1) Insert HPLC chromatograms of the analyzes;

2) Insert and discuss some mass spectra based on the reported results;

3) Improve conclusion based on the objectives in relation to mass spectrometry.

Author Response

1) Insert HPLC chromatograms of the analyzes;

We included a representative chromatogram as supplementary Fig 1.

2) Insert and discuss some mass spectra based on the reported results;

In the current study, we obtained a very small number of matching spectra due to the poor availability on the genomic information of insect species. Thus, in the discussion part, we did not discuss about the particular spectra obtained but focus and compare the similarities and dissimilarities of all obtained spectra among the samples, and also all the raw data of the mass spectra will be available online (online resource in the manuscript).

As suggested, we upload all the results of the mass spectra of the 18 samples (laboratory A and B).  The results are uploaded on line, on the Center for Computational Mass Spectrometry (UCSD);

Please see the link:  http://massive.ucsd.edu/ProteoSAFe/status.jsp?task=e6365e8315f243bba275347a54a6bdeb

The reviewer’s login is: MSV000083737_reviewer

Password: entofor_exp1_jra

If the paper will be accepted, the data will be accessible to the readers.

3) Improve conclusion based on the objectives in relation to mass spectrometry.

3) The conclusions have been changed according to the objectives.

Reviewer 3 Report

- Line 96: Check font size for text “were used”.

- Lines 103-107: Please use italics for latin names of species.

Author Response

Comments and Suggestions for Authors

- Line 96: Check font size for text “were used”.

- Lines 103-107: Please use italics for latin names of species.

Thank you, the corrections have been done.

Round 2

Reviewer 1 Report

The authors fully addressed the reviewer comments and the quality of the manuscript has significantly improved. The analytical aspects of the work have been clarified. 

My only advice would be to replace in Table 3, N.D (not determined) with 

< LoD (minor than Limit of detection).